# Bioinspired Histidine–Zn^2+^ Coordination for Tuning the Mechanical Properties of Self-Healing Coiled Coil Cross-Linked Hydrogels

**DOI:** 10.3390/biomimetics4010025

**Published:** 2019-03-18

**Authors:** Isabell Tunn, Matthew J. Harrington, Kerstin G. Blank

**Affiliations:** 1Mechano(bio)chemistry, Max Planck Institute of Colloids and Interfaces, Science Park Potsdam-Golm, 14424 Potsdam, Germany; Isabell.Tunn@mpikg.mpg.de; 2Department of Biomaterials, Max Planck Institute of Colloids and Interfaces, Science Park Potsdam-Golm, 14424 Potsdam, Germany; 3Department of Chemistry, McGill University, 801 Sherbrooke St. West, Montreal, QC H3A 0B8, Canada

**Keywords:** coiled coil, histidine–metal coordination, hydrogel, self-healing, rheology

## Abstract

Natural biopolymeric materials often possess properties superior to their individual components. In mussel byssus, reversible histidine (His)–metal coordination is a key feature, which mediates higher-order self-assembly as well as self-healing. The byssus structure, thus, serves as an excellent natural blueprint for the development of self-healing biomimetic materials with reversibly tunable mechanical properties. Inspired by byssal threads, we bioengineered His–metal coordination sites into a heterodimeric coiled coil (CC). These CC-forming peptides serve as a noncovalent cross-link for poly(ethylene glycol)-based hydrogels and participate in the formation of higher-order assemblies via intermolecular His–metal coordination as a second cross-linking mode. Raman and circular dichroism spectroscopy revealed the presence of α-helical, Zn^2+^ cross-linked aggregates. Using rheology, we demonstrate that the hydrogel is self-healing and that the addition of Zn^2+^ reversibly switches the hydrogel properties from viscoelastic to elastic. Importantly, using different Zn^2+^:His ratios allows for tuning the hydrogel relaxation time over nearly three orders of magnitude. This tunability is attributed to the progressive transformation of single CC cross-links into Zn^2+^ cross-linked aggregates; a process that is fully reversible upon addition of the metal chelator ethylenediaminetetraacetic acid. These findings reveal that His–metal coordination can be used as a versatile cross-linking mechanism for tuning the viscoelastic properties of biomimetic hydrogels.

## 1. Introduction

Histidine (His)–metal coordination is one of nature’s powerful means to mechanically reinforce biological materials, such as spider fangs, sandworm jaws or mussel byssal threads [1,2]. Mussel byssal threads are highly organized, self-assembled proteinaceous fibers. They function as a holdfast for marine mussels, such as *Mytilus edulis*, which live in the coastal intertidal zone. Byssal threads are self-assembled extracellularly in the mussel foot groove in less than 5 min [3]. The main protein building blocks comprising the tough and self-healing fibrous core of the thread, called prepepsinized collagens (preCols), are collagenous proteins containing terminal His-rich domains (HRDs). The HRDs are known to coordinate transition metal ions, such as Ni^2+^, Cu^2+^ and Zn^2+^ [4,5]. These His–metal coordination sites act as reversible cross-links, which break when dynamically loaded (e.g., by crashing waves) and dissipate high quantities of mechanical energy [6]. When allowed to rest, the coordination bonds recover, enabling self-healing of the thread [7,8]. This behavior was found to be intimately linked to the higher-order organization of intermolecular metal coordination bonds in the hierarchical structure of the byssus threads [6]. Inspired by this hierarchical structure–mechanics relationship, self-healing His–metal coordination cross-links have been implemented in star-shaped poly(ethylene glycol) (star-PEG) hydrogels as well as in supramolecular polymer films, yielding self-healing and tunable biomimetic materials [9,10,11,12,13,14,15]. The viscoelastic properties (i.e., the relaxation time) of His–metal coordinating hydrogels have been tuned using different transition metal ions as cross-linking agents [9], utilizing mixtures of different metal ions in combination [10] or by inducing in situ changes in the metal oxidation state [11,15].

In the last two decades, numerous biomimetic hybrid hydrogels have been developed, using α-helical coiled coils (CCs) as noncovalent, tunable cross-links for synthetic polymers, such as PEG [16,17,18,19,20,21]. Coiled coils are abundant protein folding motifs, frequently found in intra- and extracellular proteins with mechanical function (e.g., fibrin, vimentin and myosin), indicating that CCs can be used as versatile biomimetic cross-links [22]. Coiled coils are comprised of two to seven α-helices wrapped around each other in order to form a superhelix. The CC sequence motif consists of seven amino acid repeats (*abcdefg*), referred to as heptads, which determine folding into the superhelical structure [22]. The assembly of the CC superhelix is driven by the formation of hydrophobic contacts at positions *a* and *d* and ionic interactions at positions *e* and *g* [23]. The solvent-exposed positions *b*, c and *f* are more variable and, therefore, excellent targets for modification.

Inspired by byssal threads, we previously implemented intramolecular His–metal coordination sites into CC cross-linked star-PEG hydrogels [24]. These intramolecular metal coordination bonds were designed to strengthen individual CC cross-links by stabilizing helical turns against thermodynamic and mechanical unfolding. This strategy allowed us to reversibly tune the relaxation time of this bioinspired hydrogel [24]. For the byssal thread, however, recent investigations have suggested that intermolecular, rather than intramolecular, metal coordination bonds play an integral role in fiber self-assembly and self-healing [6]. Here, we aim to draw inspiration from natural byssal threads by rationally engineering intermolecular His–metal coordination into synthetic CC-forming peptides. Specifically, we bioengineer three intermolecular His–metal coordination sites into the solvent-exposed *f* positions of a well characterized CC heterodimer [25,26], yielding the modified CC-forming peptides A_4H3_ and B_4H3_ (Figure 1A,B). The spacing of the His residues by one heptad ensures that only intermolecular metal coordination bonds can form [27,28]. We focus on His–Zn^2+^ coordination, since it is the most abundant transition metal ion found in the HRDs of mussel byssal threads [7]. We show that the programmed intermolecular His–Zn^2+^ coordination induces the formation of higher-order α-helical assemblies. Employing the cysteine-terminated CC building block as a cross-link for star-PEG–maleimide (Figure 1C), we first prepare a hydrogel with single CC cross-links. Adding Zn^2+^, a second cross-linking mode is switched on, which triggers the formation of Zn^2+^ cross-linked aggregates inside the hydrogel.

As both the CC and the His–Zn^2+^ cross-links are reversible, this directly results in a hydrogel with two self-healing modes. Most interestingly, we demonstrate that this additional cross-linking and self-healing mode allows for reversibly and dynamically tuning the viscoelastic properties (i.e., the relaxation time) of the hydrogel as a function of the Zn^2+^ concentration.

## 2. Materials and Methods

### 2.1. Peptide Synthesis and Purification

The CC-forming peptides were synthesized using solid-phase peptide synthesis on a Tribute peptide synthesizer from Gyros Protein Technologies AB (Uppsala, Sweden). All amino acids were acquired from Bachem AG (Bubendorf, Switzerland) and 2-(6-chlor-1*H*-benzotriazol-1-yl)-1,1,3,3-tetramethylaminium-hexafluorophosphate (HCTU) was purchased from Novabiochem^®^ (Merck KGaA, Darmstadt, Germany). Dimethylformamide (DMF) and acetonitrile were obtained from VWR International GmbH (Darmstadt, Germany); and piperidine, *N*,*N*-diisopropylethylamine (DIPEA), acetic anhydride, pyridine and triflouracetic acid (TFA) were purchased from Carl Roth GmbH & Co. KG (Karlsruhe, Germany). H-rink amide ChemMatrix^®^ resin (Sigma-Aldrich Chemie GmbH, Steinheim, Germany) with a loading of 0.47 mmol per gram of resin was used to obtain peptides with an amidated C-terminus. The synthesis was performed on a 100 µM scale, using a standard Fmoc coupling protocol with 5× excess of amino acid and coupling agent (HCTU) and 12.5× excess of DIPEA at room temperature (RT). Successful coupling of the amino acids was monitored via in situ ultraviolet (UV) detection of the cleaved Fmoc group in the deprotection step. The N-terminus of the peptides was acetylated using 1:1:3 acetic anhydride:pyridine:DMF. The cleavage of the peptides was performed with 10 mL cleavage cocktail (92.5% TFA, 2.5% triisopropylsilane (98%, Sigma-Aldrich Chemie GmbH), 2.5% ethandithiol (98%, Sigma-Aldrich Chemie GmbH), 2.5% water) for 2 h at RT. The cleaved peptides were precipitated in ice-cold diethyl ether (99.5%, Carl Roth GmbH & Co. KG) and centrifuged at 15,000 *g* for 10 min at 4 °C. The precipitate was washed three times with cold diethyl ether and centrifuged as mentioned above. The crude peptides were dried under N_2_ and dissolved in 20% (A_4H3_) and 10% (B_4H3_) acetonitrile in ultrapure water, respectively.

Purification was carried out with reverse phase high-performance liquid chromatography (HPLC) (LC-20A Prominence HPLC, Shimadzu Coorporation, Kyoto, Japan), using a preparative C_18_-column (250 mm × 21 mm, Macherey-Nagel GmbH & Co. KG, Dueren, Germany). For purification, linear gradients of water (solution A: water with 0.1% TFA) and acetonitrile (solution B: acetonitrile with 0.1% TFA) were used. A gradient from 10% (B_4H3_) or 20% (A_4H3_) solution A to 100% solution B was applied over 30 min with a constant flow rate of 25 mL min^−1^. The peptides were collected based on the UV signal at 220 nm. Peptide fractions were lyophilized and subsequently used individually. The mass identity and purity of the peptides was confirmed by matrix-assisted laser desorption/ionization time-of-flight (MALDI-TOF) mass spectrometry (Autoflex Speed, Bruker, Billerica, MA, USA), using 2,5-dihydrobenzoic acid (DHB) (Sigma-Aldrich Chemie GmbH) as the matrix (Appendix A).

### 2.2. Circular Dichroism Spectroscopy

Circular dichroism (CD) spectra were recorded to investigate the secondary structure of the individual CC-forming peptides and the CC, both in the absence and presence of ZnCl_2_. As a negative control, CaCl_2_ was used, which is known to have only a very weak interaction with His [30]. Peptide stock solutions were prepared in ultrapure water at a concentration of 5 mg mL^−1^. The stock solutions were mixed in a 1:1 molar ratio und subsequently diluted in piperazine-1,4-bis(propanesulfonic acid) (PIPPS, Merck KGaA) buffer (10 mM PIPPS at pH 8.1, 137 mM NaCl, 2.7 mM KCl) to obtain a final CC concentration of 30 µM. For measurements in the presence of metal ions (M^2+^), ZnCl_2_ or CaCl_2_ (Alfa Aesar, Karlsruhe, Germany) was added to the peptide solution (1:1 M^2+^:His) before diluting with PIPPS buffer. Then, 50 mM tris(2-carboxyethyl)phosphine (TCEP) (Thermo Fisher Scientific, Waltham, MA, USA) was added to a concentration of 300 µM to reduce possible disulfide bonds. The CD spectra were acquired with a Chirascan qCD spectrometer (Applied Photophysics Ltd., Leatherhead, UK) using a quartz cuvette (Hellma GmbH & Co. KG, Müllheim, Germany) with a path length of 1 mm. All spectra were recorded from 200 to 250 nm with a bandwidth of 1 nm and a step resolution of 1 nm. The spectra were recorded at RT with an integration time (time per point) of 1 s nm^−1^ and three accumulations, which were averaged. The buffer spectrum was recorded for baseline subtraction using identical parameters. The molar ellipticity [θ] (deg cm^2^ dmol^−1^) was calculated according to Equation (1):(1)$$\left[\theta \right]=\frac{100\cdot \theta }{c\cdot d}$$with the ellipticity θ (deg), the total peptide concentration *c* (M) and the path length *d* (cm) [31].

### 2.3. Confocal Raman Spectroscopy

The CC A_4H3_B_4H3_ was investigated with confocal Raman spectroscopy using peptide films. The peptides A_4H3_ and B_4H3_ were dissolved in ultrapure water to a concentration of 5 mg mL^−1^, mixed in a 1:1 molar ratio, and then diluted to yield 1 mM A_4H3_B_4H3_. To investigate His–metal coordination, 20 mM ZnCl_2_ or CaCl_2_ was added to obtain a final concentration of 3 mM (1:1 M^2+^:His). The pH was raised to 8 using 1 µL of 50 mM NaOH (Carl Roth GmbH & Co. KG). The solution (7 µL) was dried on a quartz glass coverslip (TED Pella, Inc., Redding, Canada). A confocal Raman microscope (alpha300 R, WITec, Ulm, Germany) was used for the measurements. The microscope was equipped with a 20× objective (NA 0.4, Nikon, Tokyo, Japan) and a piezo scanner (P-500, Physik Instrumente GmbH & Co. KG, Karlruhe, Germany). The sample was measured with a linearly polarized laser (λ = 532 nm; Oxxius SA, Lannion, France) with a polarization angle of 0° and no analyzer in the light path. A thermoelectrically cooled charge-coupled device (CCD) detector (DU401A-BV, Andor, Belfast, UK) with an integration time of 10 s and six accumulations was used to detect the Raman scattered light. Spectra from three different positions on the sample were collected and averaged. The measurement and subsequent data analysis were performed with the software ScanCtrl Spectroscopy Plus (Version 1.38, WITec) and Project FOUR (Version 4.1, WITec). OPUS 7.0 (Bruker) was used for baseline correction (rubberband method, linear, 1 pt) and smoothing (based on the Savitzky–Golay algorithm, 9 pt). The spectra were normalized with respect to the maximum of the amide I peak (Origin Pro 2015 b9.2.257, OriginLab Corporation, Northampton, MA, USA).

### 2.4. Hydrogel Preparation

The cysteine-terminated peptides A_4H3_ and B_4H3_ were used to cross-link maleimide-functionalized 40 kDa star-shaped poly(ethyleneglycol) (star-PEG–maleimide) (JenKem Technology Co., Ltd., Beijing, China). The hydrogels were prepared as follows: First, the individual peptides were dissolved in phosphate-buffered saline (PBS) (10 mM Na_2_HPO_4_/2 mM KH_2_PO_4_ at pH 7.4, 137 mM NaCl, 2.7 mM KCl (all reagents from Carl Roth GmbH & Co. KG)) in a concentration of 10 mg mL^−1^. Second, possible disulfide bonds were reduced with Pierce™ immobilized TCEP disulfide reducing gel (Thermo Fisher Scientific) for 1.5 h at 4 °C, while mixing (2000 rpm), to ensure that all cysteines are available for the reaction with star-PEG–maleimide. The concentration of free cysteines was determined with Ellman’s reagent (Thermo Fisher Scientific). Third, the reduced peptides were coupled to star-PEG–maleimide (separately to yield star-PEG–A_4H3_ and star-PEG–B_4H3_ (Figure 1C). To ensure that all arms of star-PEG–maleimide are functionalized, a 1.2-fold excess of thiol groups as determined with Ellman’s reagent was used for each reaction. The reaction mixture was incubated for 15 min (RT, 800 rpm) and the excess of peptide was removed using ultrafiltration (molecular weight cut-off 10 kDa, Amicon^®^ Ultra, Merck KGaA). The buffer was changed to ultrapure water during ultrafiltration (5×, 14,000 *g*, 10 min). The purified star-PEG–peptide conjugates were lyophilized and stored at −20 °C. For hydrogel preparation, the star-PEG–peptide conjugates were dissolved in PIPPS buffer (pH 8.1) to a concentration of 0.5 mM. star-PEG–A_4H3_ and star-PEG–B_4H3_ were mixed in a 1:1 ratio and hydrogels formed in less than 1 min. Air bubbles entrapped in the hydrogel were removed by centrifugation (2 min, 2000 *g*). To study the effect of metal coordination, ZnCl_2_ or CaCl_2_ (negative control) was added from a 100 mM stock solution in different M^2+^:His ratios right after forming the hydrogel and mixed thoroughly. The reversibility of Zn^2+^ coordination was investigated, adding the chelating agent ethylenediaminetetraacetic acid (EDTA) (Carl Roth GmbH & Co. KG) at a final concentration of 25 mM. The excess EDTA (4× with respect to the His concentration) competes with His for Zn^2+^ and is thus expected to disrupt the Zn^2+^–His interaction. All hydrogels were incubated at 4 °C for at least 4 h before the measurement to allow equilibration.

### 2.5. Oscillatory Shear Rheology

Hydrogels were characterized with rheology (MCR 302 and MCR 301, Anton Paar GmbH, Graz, Austria), using a 12 mm diameter cone-plate geometry (CP12-1, angle 1°, gap width 0.02 mm, Anton Paar GmbH). To prevent evaporation, the rheometer was equipped with a temperature-controlled hood (25 °C). The linear viscoelastic range was determined in amplitude sweeps, using strain amplitudes ranging from 0.1% to 1000% and vice versa. During the amplitude sweeps, the frequency was kept constant at 10 rad s^−1^. After a short resting time of 2 min, a frequency sweep was conducted from 100 to 0.001 rad s^−1^ (hydrogels with metal ions) or 0.005 rad s^−1^ (hydrogel without metal ions and with 1:1 Ca^2+^:His). During the frequency sweeps, the strain amplitude was kept constant at 1%, which lies in the linear viscoelastic range. The frequency sweeps were used to obtain information about the dynamic viscoelastic properties of the hydrogels in the presence and absence of metal ions. In the frequency sweeps, the relaxation time of noncovalent cross-links can be determined from the inverse of the crossover frequency *ω*_c_ (*τ*_fs_ = 1/*ω*_c_) of the storage (*G*’) and loss (*G*”) moduli. Using different hydrogel samples, three independent frequency sweeps were performed to obtain the mean of the relaxation time and the corresponding standard error of the mean (SEM).

Additionally, stress relaxation experiments were performed. A step-strain of 10% was applied and the relaxation modulus *G*(*t*) was monitored over time. The stress relaxation behavior of many noncovalently cross-linked, polymer-based hydrogels can be described with Kohlrausch’s stretched exponential relaxation model (Equation (2)):(2)$$G\left(t\right)=G_{0}\;exp(-{\left(t/\tau_{sr}\right)}^{\alpha })$$with the relaxation time *τ*_sr_ (s), the initial plateau modulus *G*_0_ (Pa) and the fitting parameter *α*, which is governed by the physical properties of the material [32,33]. The fit was performed with the Levenberg–Marquardt iteration mechanism in Origin Pro 2015 b9.2.257. *G*_0_ was fixed to the value of *G’* in the linear viscoelastic region of an amplitude sweep which was performed with the same hydrogel sample before the stress relaxation experiment.

## 3. Results

### 3.1. Secondary Structure and Metal Coordination Mode of the His-Modified Coiled Coil

Inspired by the self-healing capability and the hierarchical architecture of byssal threads, we rationally engineered a reversibly switchable second cross-linking mode into a CC cross-linked PEG hydrogel. This cross-linking mode uses His–metal coordination bonds to mediate the formation of intermolecular interactions between CC-forming peptides. The His–metal coordination sites were rationally designed into the solvent-exposed *f* positions of a well characterized heterodimeric CC (Figure 1) [25,26]. The spacing between His residues in A_4H3_B_4H3_ is one heptad, which corresponds to 10.8 Å along the helical axis [27]. The optimum length of a coordination bond between the metal ion and the imidazole side chain of His is 2.1 Å [28]. Thus, the spacing of the His residues prohibits intramolecular His–metal coordination within a folded CC molecule. In this study, we focused on the transition metal ion Zn^2+^, since His–Zn^2+^ coordination is frequently found in byssal threads [7].

To investigate the ability of the His residues to coordinate Zn^2+^ and its effect on the secondary structure of the CC, we used CD and Raman spectroscopy. In the absence of metal ions, the CD spectrum of A_4H3_B_4H3_ shows a typical α-helical structure with two minima at 208 and 222 nm (Figure 2A). The ratio between the minima (R_222/208_) is 1.10 and indicates a well-defined CC structure [34,35]. When Zn^2+^ was added in a ratio of 1:1 Zn^2+^:His, the intensity of the minimum at 208 nm decreased (R_222/208_ = 1.71) and the peak at 222 nm shifted to longer wavelengths (225 nm). At the same time, the absolute molar ellipticity decreased and the solution became slightly opaque, suggesting aggregation and precipitation in the sample. Similar CD spectra have previously been reported for α-helical nanofibers consisting of multiple CCs [36,37] as well as nonfibrillar α-helical aggregates [38,39,40]. Based on these earlier reports and the appearance of a visible precipitate, we conclude that A_4H3_B_4H3_ forms extended α-helical assemblies upon adding Zn^2+^. In the presence of Ca^2+^, which interacts with His residues very weakly [30], A_4H3_B_4H3_ maintained a typical α-helical CC conformation. The CD spectrum does not show any sign of extended α-helical assemblies. Instead, R_222/208_ decreases to 0.97 and the minimum at 208 nm shifts to 206 nm. This is characteristic of a small destabilization of the CC structure, which may arise from the interaction of Ca^2+^ with glutamic acid (Glu) residues in A_4H3_, thus, weakening the ionic interactions in the *e* and *g* positions. In the case of the individual peptides (A_4H3_ and B_4H3_), a random coil structure was seen in the absence of metal and in the 1:1 Ca^2+^:His control sample, whereas the signature of α-helical assemblies was observed in the presence of 1:1 Zn^2+^:His (Appendix A). This suggests Zn^2+^-induced folding and assembly of the individual peptides. It is thus not possible to unambiguously conclude if the α-helical assemblies in the A_4H3_B_4H3_ sample resemble Zn^2+^ cross-linked CC clusters or rather aggregated mixtures of the individual peptides. The second possibility would require the Zn^2+^-induced dissociation of a specific and thermostable CC structure (*T*_m_ = 64.7 ± 1.4 °C), however, which we consider to be highly unlikely.

Raman spectroscopy was used to confirm the secondary structure of the CC and to investigate His–Zn^2+^ coordination in more detail. The amide I peak of A_4H3_B_4H3_ at 1651 cm^−1^ (Figure 2B) confirms the α-helical secondary structure [41]. In the presence of Zn^2+^, the amide I peak is shifted slightly to 1654 cm^−1^, which is well within the spectral range typically assigned to α-helices. In addition, a small shoulder appears at 1680 cm^−1^. This shoulder was previously observed in keratin, the CC-based building block of hair, which is predominantly α-helical. These amide I shoulders, which appeared at 1677 cm^−1^ or 1685 cm^−1^, were variably assigned as β-sheet or unordered structures [42,43]. We propose that this small and measurable effect on the protein secondary structure, observed for the Zn^2+^-coordinated and aggregated CC, originates from a distortion of the helices upon aggregation. This appears consistent with the results from CD spectroscopy, where a shift of the 222 nm peak and an altered R_222/208_ ratio suggests a different chiral environment of the peptide bond. It is notable that the presence of Ca^2+^ does not lead to a shift of the amide I band and only to a minor increase of the shoulder at 1680 cm^−1^.

Changes in the protonation and coordination state of His directly affect the vibrational modes of the C_4_=C_5_ bond of the imidazole ring in the range of 1550–1640 cm^−1^ [44]. Free, deprotonated His at pH 8 shows a characteristic peak at 1574 cm^−1^. In the presence of 1:1 Zn^2+^:His, two metal coordination peaks are observed at 1556 and 1605 cm^−1^, indicating the presence of two populations of metal-bound His residues. In one population, both nitrogen atoms of the imidazolium moiety simultaneously coordinate one Zn^2+^ ion (i.e., bridging). In the other population, Zn^2+^ is bound to a single nitrogen atom (Figure 2C). The bridging coordination mode was also reported for His–Zn^2+^ coordinating β-sheet-forming peptides, which were derived from the HRDs of byssal threads [45]. In contrast to Zn^2+^, the addition of Ca^2+^ does not lead to a shift of the peak at 1574 cm^−1^ while a very small peak appears at 1594 cm^−1^. This indicates that most of the His residues are free and only a small amount of Ca^2+^ is weakly coordinated. These results are in agreement with the CD spectra. Overall, the spectroscopic characterization shows that His–Zn^2+^ coordination leads to the formation of predominantly α-helical assemblies, whereas Ca^2+^ is not able to mediate these higher-order interactions.

### 3.2. Switching the Coiled Coil Cross-Linked Hydrogel from Viscoelastic to Elastic Using His–Zn^2+^ Coordination

Although the formation of hierarchical, self-healing structures is a principle often found in nature, the implementation of tunable and reversible higher-order assemblies in biomimetic hydrogels remains challenging. To investigate if the formation of intermolecular His–Zn^2+^ coordination bonds enables us to influence the emergent mechanical properties of CC cross-linked star-PEG hydrogels, we coupled the peptides A_4H3_ and B_4H3_ to star-PEG–maleimide, utilizing terminal cysteines introduced at the N-terminus of A_4H3_ and the C-terminus of B_4H3_ (see Figure 1C and Section 2.4). The hydrogels were formed in less than 1 min when mixing star-PEG–A_4H3_ and star-PEG–B_4H3_ in a stoichiometric ratio. ZnCl_2_ or CaCl_2_ were added to the formed hydrogels in different M^2+^:His ratios to study the effect of these metal ions on the hydrogel properties. Interestingly, upon addition of ZnCl_2_ the hydrogel expelled some water, which was not observed when CaCl_2_ was added. This is a first indication of a change in the cross-linking mode of the hydrogel.

Oscillatory shear rheology was used to characterize the CC cross-linked hydrogels. Amplitude sweeps, covering strain amplitudes from 0.1% to 1000%, show that the CC cross-linked hydrogels possess a large linear viscoelastic range in the absence of metal ions as well as in the presence of 1:1 Ca^2+^:His (Figure 3A,B). The storage modulus *G*’ remains higher than the loss modulus *G*” up to strain amplitudes of approximately 100%. At strain amplitudes above 100%, a decrease of *G*’ is accompanied by an increase of *G*”, which indicates the onset of cross-link failure [46]. When the strain amplitude is gradually lowered to 0.1%, the original viscoelastic properties recover. The noncovalent CC cross-links rapidly reform and the material self-heals even after repeating the amplitude sweep four times (Appendix A). In the presence of 1:1 Zn^2+^:His, the initial *G*’ is four times higher than for the hydrogel with no metal ions and the linear viscoelastic range is reduced (0.1–10%) (Figure 3C). Moreover, in contrast to hydrogels without metal ions, the Zn^2+^-fortified hydrogel exhibits a different self-healing behavior. Only ≈80% of *G*’ is recovered immediately when the strain amplitude is reduced to 0.1%. This reveals that the complete reformation of Zn^2+^ cross-linked aggregates is not instantaneous. Repeating the amplitude sweep three more times revealed that *G*’ remains at ≈80% of the initial *G*’ (Appendix A). When the hydrogel is allowed to rest for 1 h at 25 °C, ≈90% of the initial *G*’ is recovered. In combination, this suggests that a certain, constant fraction of intermolecular cross-links can form quickly. The further growth of Zn^2+^ cross-linked aggregates proceeds more slowly, however, showing that the hydrogels possess time-dependent self-healing dynamics comparable to the His–metal fortified mussel byssal threads [7]. Adding an excess of the metal chelator EDTA, which competes with His for Zn^2+^ binding, we show that the effect of His–Zn^2+^ coordination is fully reversible. Following addition of EDTA, the amplitude sweep recovers the same characteristic shape as in the absence of metal ions (Figure 3D).

Frequency sweeps performed at a strain amplitude of 1% provide information about the dynamic mechanical properties of the hydrogels. According to the Maxwell model [10], the characteristic relaxation time *τ*_fs_ of noncovalently cross-linked hydrogels can be obtained from the crossover frequency *ω*_cr_ of *G*’ and *G*” (*ω*_cr_ = 1/*τ*_fs_). In the absence of metal ions, the hydrogel behaves according to the Maxwell model with a *τ*_fs_ of 11.2 ± 4.0 s (mean ± SEM, *n* = 3; Figure 4A and Appendix A). In the presence of 1:1 Zn^2+^:His, the hydrogel displays no crossover of *G*’ and *G*”, revealing purely elastic properties over the frequency range measured. Thus, we were able to switch the properties of the CC-based hydrogels from viscoelastic to elastic-like using intermolecular His–Zn^2+^ coordination as a second cross-linking mode. In contrast, hydrogels containing 1:1 Ca^2+^:His possess a relaxation time of 4.7 ± 0.5 s, which is highly similar to the relaxation time of the metal-free hydrogel (Figure 2B). After addition of excess EDTA to the Zn^2+^-containing hydrogel, the relaxation time shifted back to 5.9 ± 0.7 s. This proves that the formation of Zn^2+^ cross-linked aggregates is fully reversible. Cross-linking is thus based on single CCs in the absence of metal ions, while the addition of Zn^2+^ switches on intermolecular metal coordination as a second cross-linking mode (Figure 4B). The formation of Zn^2+^ cross-linked aggregates causes a significant change in the mechanical properties of the hydrogel, including the transition from viscoelastic to elastic-like and time-dependent self-healing, which resembles the mussel byssus.

### 3.3. Tuning the Relaxation Time of the Coiled Coil Cross-Linked Hydrogel with Different Zn^2+^:His Ratios

To test if the relaxation time of the CC cross-linked hydrogel can be precisely tuned with His–Zn^2+^ coordination, we assessed the mechanical response of the hydrogels using different Zn^2+^:His ratios. At a ratio of 1:1 Zn^2+^:His, the hydrogel behaves like an elastic solid in the frequency range measured with no crossover of *G*’ and *G*”. At ratios of 1:2 and 1:10 Zn^2+^:His (Figure 5), the hydrogel largely maintains its elastic properties and a small maximum is seen for *G*” at 0.02 rad s^−1^. When the Zn^2+^:His ratio is further lowered to 1:20, a crossover of *G*’ and *G*” is detected, resulting in *τ*_fs_ = 51.8 s. Lowering the Zn^2+^:His ratio to 1:100 yields a relaxation time of 10 s, which is close to the value observed in the hydrogel without metal ions (*τ*_fs_ = 11.2 s). The His–Zn^2+^ coordination-mediated transition between elastic-like and viscoelastic clearly depends on the number and size of Zn^2+^ cross-linked aggregates, which can be controlled by a gradual stepwise decrease in the Zn^2+^ concentration, as revealed in further frequency sweeps at ratios of 1:5, 1:15 and 1:50 Zn^2+^:His (Appendix A). Amplitude sweeps (Appendix A) show that also the linear viscoelastic range of all hydrogels is affected by the altered Zn^2+^ concentration, but much less than the relaxation time.

As already observed in the CD spectra, the individual peptides A_4H3_ and B_4H3_ also form α-helical assemblies in the presence of Zn^2+^ (1:1 Zn^2+^:His ratio). This leads to the formation of hydrogels when adding Zn^2+^ to the individual star-PEG–A_4H3_ and star-PEG–B_4H3_ conjugates. However, frequency sweeps at different Zn^2+^:His ratios reveal a different concentration dependence (Appendix A). When lowering the concentration of Zn^2+^, star-PEG–A_4H3_ forms hydrogels down to a Zn^2+^:His ratio of 1:10, whereas star-PEG–B_4H3_ forms a weak hydrogel only at 1:1 Zn^2+^:His and is liquid at lower Zn^2+^ concentrations. The ability of star-PEG–A_4H3_ to form a hydrogel in the absence of star-PEG–B_4H3_ most likely originates from the cooperative coordination of the three His and the eight Glu residues of A_4H3_ (Figure 1A). It seems plausible that Zn^2+^ coordination might induce α-helical folding of the peptide, an effect which was previously described by Aili et al. [47] for a His- and Glu-rich helix-loop-helix motif. Based on these observations, we conclude that the CC strongly contributes to the overall viscoelastic properties of the hydrogel and that the addition of Zn^2+^ induces the hierarchical assembly of CCs into higher-order structures.

To obtain additional information about the presence and timescales of relaxation processes in the A_4H3_B_4H3_-containing hydrogels with a 1:1, 1:2 or 1:10 Zn^2+^:His ratio, stress relaxation experiments were performed. The relaxation times were not accessible in the frequency sweeps, as the measurement time could not be further extended without significant drying of the hydrogel samples. For the stress relaxation experiments, the hydrogels were exposed to a step-strain of 10% and the relaxation of the hydrogel properties was monitored as a function of time. Figure 6 shows that without metal ions, the hydrogel fully relaxes (*G*(*t*) = 0) over a timescale of 100 s. In contrast, the hydrogel containing 1:20 Zn^2+^:His requires 10,000 s to relax and the hydrogels with 1:1 and 1:10 Zn^2+^:His do not fully relax on the timescale of the experiment. These results are in agreement with the frequency sweeps, showing that increasing the Zn^2+^:His ratio increases the relaxation time of the hydrogel.

The stress relaxation curves were fitted to Kohlrausch’s stretched exponential relaxation model (Equation (2), Appendix A). In the absence of metal ions, the relaxation times obtained from the frequency sweep (*τ*_fs_ = 11.2 s) and the stress relaxation experiment (*τ*_sr_ = 6.6 s) are highly similar (Appendix A). In the presence of 1:20 Zn^2+^:His, the relaxation time determined from fitting the stress relaxation curve is 34.4 s, which is in the same order of magnitude as *τ*_fs_ (51.8 s). Thus, both methods yield comparable relaxation times. In the presence of 1:10 and 1:1 Zn^2+^:His, *τ*_sr_ is 1244 s and 1272 s, respectively, revealing that the hydrogel containing 1:1 or 1:10 Zn^2+^:His relax almost equally slow.

Kohlrausch’s stretched exponential relaxation model is often used for materials with physically constrained cross-links and the shape of the fit is governed by the parameter *α*. Here, the parameter *α* decreases from 0.59 for the hydrogel without metal ions to 0.24 for the hydrogel containing 1:1 Zn^2+^:His. Currently, it is not fully understood how *α* is related to the hydrogel relaxation processes on a molecular level. One current hypothesis is that *α* = 1 if the hydrogel has only one distinct relaxation time (single exponential) and α decreases when a superposition of exponentially relaxing processes is present (i.e., when the cross-links possess a distribution of relaxation times) [48]. Another hypothesis is that relaxation at the molecular level is always stretched exponential because of cooperative molecular motions and entanglements in the hydrogel network [48].

According to the first hypothesis, *α* should be 1 in the CC cross-linked hydrogel without metal ions (single CC cross-links with a distinct *τ*_sr_); however, we observe a value for *α* of 0.59. This indicates that cooperative molecular motions and entanglements might play a role in the hydrogel relaxation process even when only highly defined CC cross-links are present. Increasing Zn^2+^:His ratios lead to a decrease of *α* to 0.24. Intermolecular His–Zn^2+^ coordination induces the formation of α-helical assemblies with varying size and number of metal coordination bonds. Clearly, this causes a broad distribution of timescales for the relaxation of these Zn^2+^ cross-linked aggregates and at the same time increases the probability for cooperative motions and entanglements. The relaxation and reformation of aggregates is much more restricted by the length of the individual PEG chains and the network architecture than it is the case for individual CC cross-links. The current interpretation thus assumes that both processes influence the *α* parameter. Overall, the results reveal that His–Zn^2+^ coordination is a powerful means to induce the formation of higher-order assemblies and to dynamically tune the relaxation time of the hydrogel over nearly three orders of magnitude.

## 4. Discussion

Inspired by the self-healing ability of mussel byssal threads, we rationally engineered intermolecular His–metal coordination sites into a well characterized CC. We were able to utilize this programmed second cross-linking mode to induce the reversible formation of Zn^2+^ cross-linked aggregates. Aggregation causes a dramatic change in the mechanical properties of the hydrogel, which we attribute to a progressive growth of His–Zn^2+^ coordinating α-helical assemblies, which is controlled by the Zn^2+^ concentration. Over the frequency range tested, elastic-like properties of the hydrogel were ultimately obtained in the high concentration range (Zn^2+^:His ratios of 1:1 and 1:2) (Figure 7). Furthermore, in the presence of 1:1 Zn^2+^:His, *G*’ is four times higher when compared to the hydrogel without metal ions, which indicates a higher degree of cross-linking (Figure 4) [49]. This is further proof that switching on His–Zn^2+^ coordination as a second cross-linking mode alters the network connectivity and therefore has a direct impact on the relaxation time of the hydrogel. A similar reinforcement of material properties was observed when adding Zn^2+^ to a different type of bioinspired hydrogel, which was derived from His-rich proteins found in the jaw of the marine worm *Nereis virens* [2,50]. Switchable viscoelastic properties have been reported previously for a catechol-functionalized star-PEG hydrogel, which was viscoelastic when cross-linked through single tris-coordinated catechol–Fe^3+^ complexes and transitioned into an elastic-like material when using Fe_3_O_4_ nanoparticles as cross-links [12]. Changing the size of the nanoparticles and thereby the number of polymer chains forming a cross-link, Li et al. [12] were also able to shift the relaxation time of the hydrogel. 

Here, we implement His–Zn^2+^ coordination sites into CC-forming peptides as an additional cross-linking mode, thereby inducing the formation of His–Zn^2+^ coordinating, α-helical assemblies. This is in contrast to the catechol-functionalized hydrogel, which relies only on the catechol–Fe^3+^ interaction and uses nanoparticle size to tune cross-link size and stoichiometry. Only the implementation of two cross-linking modes allows for fine tuning network connectivity while maintaining an intact hydrogel network. Metal ion-induced cross-linking thus provides us with the possibility to reversibly switch the properties of the hydrogel from viscoelastic (no metal) to elastic-like (Zn^2+^:His ratio of 1:1) and back to viscoelastic (addition of EDTA). More importantly, increasing the Zn^2+^:His ratio leads to a higher number of chains involved in the formed Zn^2+^ cross-linked aggregates, which provides gradual tunability of the relaxation time, using Zn^2+^ (and EDTA) as a soluble additive to alter hydrogel properties (Figure 7).

The hydrogel revealed a time-dependent self-healing behavior after inducing the formation of Zn^2+^ cross-linked aggregates. In contrast, hydrogels cross-linked solely via single CCs (no metal) showed that the initial *G*’ recovers rapidly, when releasing the strain in an amplitude sweep. This is based on the fast association of single CCs in the absence of mechanical load. In contrast, if His–Zn^2+^ coordination is switched on as the second cross-linking mode, *G*’ remains at only ≈80% of the initial value when the strain amplitude is gradually reduced to 0.1%. This suggests that only small aggregates assemble on the timescale of the experiment, while larger aggregates need additional time to reform. After healing for 1 h at 25 °C, *G*’ recovers to ≈90% of the initial value. With increasing size, the association dynamics becomes consecutively slower as the cross-linked network hinders chain diffusion and aggregate growth. Also, the stress relaxation experiments confirm this observation. The stretched exponential behavior with a small *α* of 0.24 (1:1 Zn^2+^:His) suggests a healing process that spans timescales of many orders of magnitude. Remarkably, this time-dependent response is reminiscent of mussel byssal threads, which require extended rest periods to recover up to 90% of their initial material stiffness and toughness [51,52]. The self-healing ability of byssal threads is mainly attributed to the reformation of a complex intermolecular network of His–metal coordination bonds in the HRDs of preCol building blocks. These surround a β-sheet framework that provides reversible hidden length when the sacrificial protein–metal bonds break [6]. Thus, self-healing of our Zn^2+^ cross-linked hydrogel resembles the natural model system, the byssal thread.

Adding an excess of EDTA to remove the metal ions and to dissociate the Zn^2+^ cross-linked aggregates, we demonstrate that the second cross-linking mode and, thus, the relaxation time can be reversibly switched on and off. The availability of two cross-linking modes is a clear advantage of the CC-based, Zn^2+^ cross-linked hydrogel when compared to simpler mussel-inspired hydrogels. These are solely based on His–metal cross-links and completely dissolve in the presence of EDTA. To the best of our knowledge, peptide–polymer hybrid hydrogels with comparably tunable hierarchical structure and mechanical properties have not been reported previously. Using other divalent transition metal ions, such as Ni^2+^, Cu^2+^ or Co^2+^ might further increase the tunable range of the mechanical properties of this CC-based hydrogel [9,10,53]. In summary, the presented work highlights that His–metal coordination is a powerful means for introducing a switchable, second cross-linking mode into self-healing biomimetic hydrogels. It facilitates reversible self-assembly and, as a direct result, allows for controlling the hydrogel relaxation time.

## Figures and Tables

**Figure 1 biomimetics-04-00025-f001:**
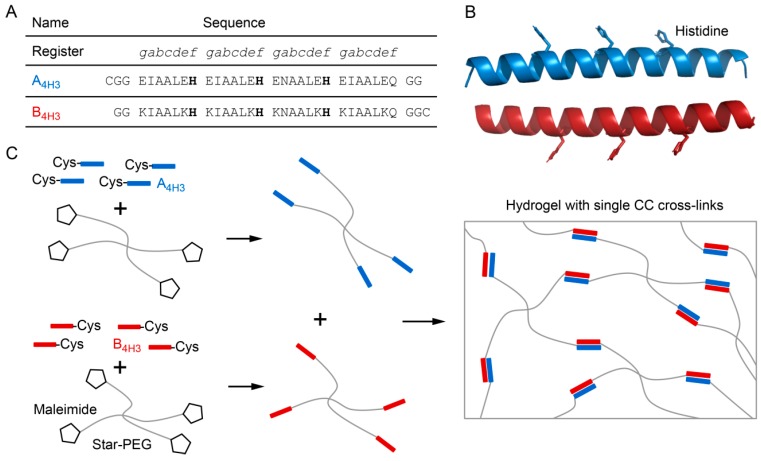
Metal coordinating coiled coil (CC) cross-linked hydrogel. (**A**) Sequences of the acidic (A_4H3_) and basic (B_4H3_) CC-forming peptides, carrying three histidines (bold) in the solvent-exposed *f* positions. Cysteine residues were added at the N-terminus of A_4H3_ and the C-terminus of B_4H3_ to facilitate coupling to maleimide-functionalized star-shaped poly(ethylene glycol) (star-PEG–maleimide). (**B**) Structure of the CC constructed in CCBuilder 2.0 [29]. The side chains of the histidine residues are highlighted. (**C**) Schematic representation of the hydrogel preparation and structure. A_4H3_ and B_4H3_ are separately coupled to star-PEG–maleimide via their terminal cysteine residues. The resulting PEG–peptide conjugates are then mixed in a 1:1 ratio to obtain a CC cross-linked hydrogel.

**Figure 2 biomimetics-04-00025-f002:**
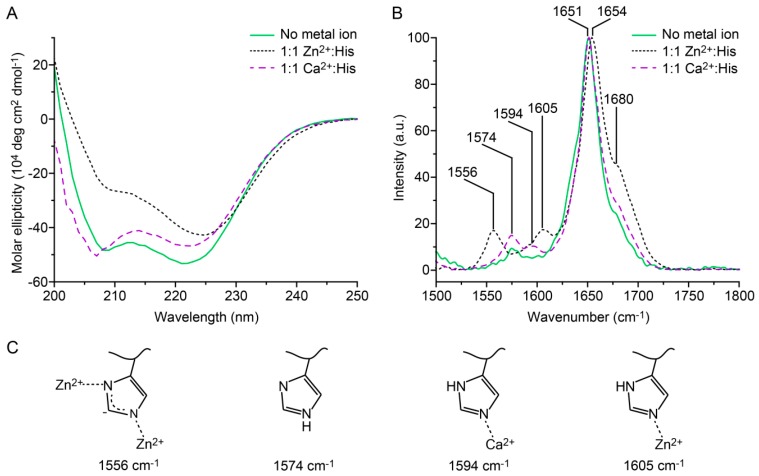
Circular dichroism and Raman spectroscopy of A_4H3_B_4H3_. (**A**) Circular dichroism spectra measured in noncoordinating PIPPS buffer. (**B**) Raman spectroscopy performed on dried thin films of the coiled coil (CC). With both techniques, the CC was studied in the absence of metal ions and in the presence of 1:1 Zn^2+^:His or 1:1 Ca^2+^:His (negative control). (**C**) Scheme of the protonation and coordination states of His observed in Raman spectroscopy. The amide I peaks at 1651 cm^−1^ and 1654 cm^−1^ indicate an α-helical secondary structure. The assignment of the peak at 1680 cm^−1^ is less clear and can indicate the existence of β-sheet or unordered secondary structures [41,42,43].

**Figure 3 biomimetics-04-00025-f003:**
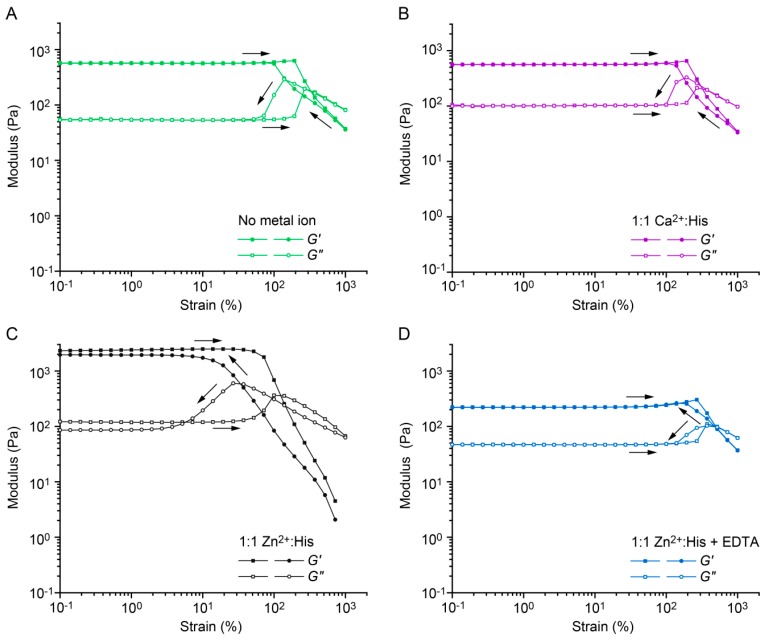
Amplitude sweeps of A_4H3_B_4H3_-containing hydrogels in PIPPS buffer. Amplitude sweeps were performed at a constant angular frequency of 10 rad s^−1^, changing the strain amplitude from 0.1% to 1000% and vice versa. (**A**) In the absence of metal ions; (**B**) in the presence of 1:1 Ca^2+^:His; (**C**) 1:1 Zn^2+^:His; (**D**) or 1:1 Zn^2+^:His + 25 mM EDTA (4× excess with respect to the His concentration).

**Figure 4 biomimetics-04-00025-f004:**
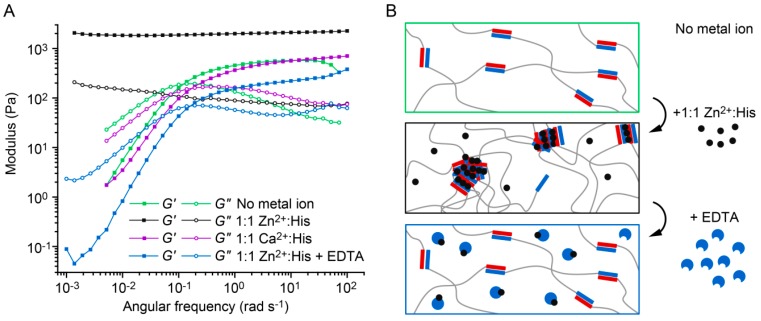
Frequency sweeps of A_4H3_B_4H3_-containing hydrogels in PIPPS buffer. (**A**) Frequency sweeps were performed in the absence of metal ions, in the presence of 1:1 Ca^2+^:His, 1:1 Zn^2+^:His and 1:1 Zn^2+^:His + 25 mM EDTA. The frequency sweeps were performed over a range of 100 to 0.001 rad s^−1^ at a constant strain amplitude of 1%. (**B**) Schematic representation of the cross-linking mode in hydrogels without metal ions, with 1:1 Zn^2+^:His and after adding EDTA.

**Figure 5 biomimetics-04-00025-f005:**
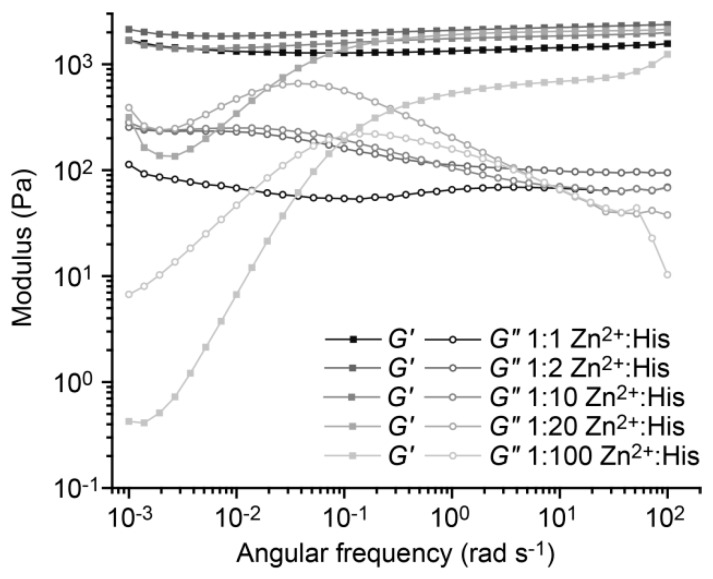
Frequency sweeps of A_4H3_B_4H3_-containing hydrogels with different Zn^2+^:His ratios. The frequency sweeps were performed from 100 to 0.001 rad s^−1^ at a constant strain amplitude of 1%, using Zn^2+^:His in ratios of 1:1, 1:2, 1:10, 1:20 and 1:100.

**Figure 6 biomimetics-04-00025-f006:**
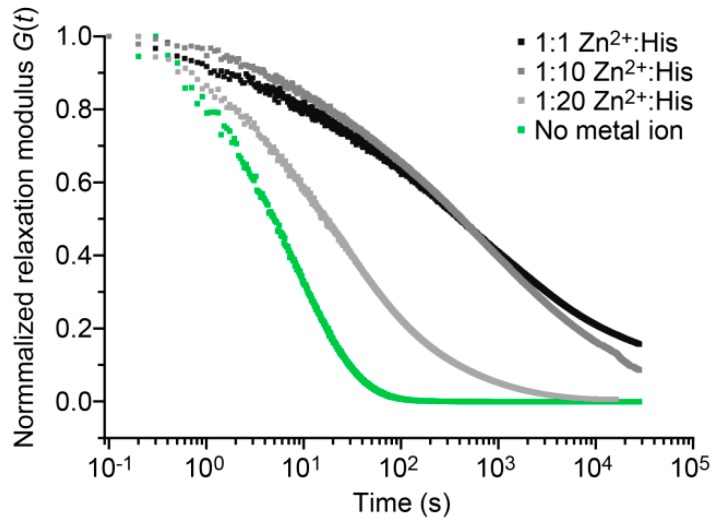
Stress relaxation of A_4H3_B_4H3_-containing hydrogels without metal ions and with different Zn^2+^:His ratios (1:1, 1:10, and 1:20). The stress relaxation of the hydrogels was monitored after applying a step-strain of 10%.

**Figure 7 biomimetics-04-00025-f007:**
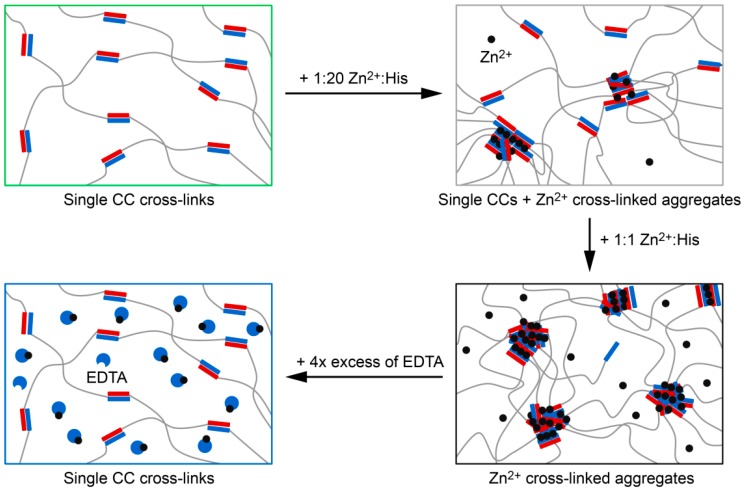
Scheme of the cross-linking modes in the hydrogel. In the absence of metal ions, only single coiled coil (CC) cross-links form. At 1:20 Zn^2+^:His, the hydrogel is cross-linked by a mixture of single CCs and small Zn^2+^ cross-linked aggregates. At 1:1 Zn^2+^:His most of the CC-forming peptides participate in these aggregates. The formation of Zn^2+^ cross-linked aggregates is fully reversible when EDTA is added.

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
