# Peer review of "Bioinspired Histidine–Zn2+ Coordination for Tuning the Mechanical Properties of Self-Healing Coiled Coil Cross-Linked Hydrogels"

_biomimetics, 2019, doi:10.3390/biomimetics4010025_

Round 1
Reviewer 1 Report
This paper presents an experimental study on 4-arm PEG hydrogels crosslinked initially by alpha helical coiled coils and then additionally by ionic metal interactions between divalent cations and HIS domains. The gels transform from linear viscoelastic at small strains without metal crosslinking to linear elastic at small strains with metal crosslinking. The experimental methodology, choice of controls, and theoretical analyses are sound and reasonably interpreted by the authors. This topic is of interest to the growing research community investigating and trying to use tunable gels. Additionally, this manuscript is well organized and clearly written. I have just a few small suggestions for editing, these are listed below.
The authors show and discuss the experimental effect of adding EDTA in the results, but do not explain that the EDTA is expected to bind the Zn2+ preferentially over the HIS until the discussion. This experiment does not make sense unless the reader understands the point of the EDTA, so as sentence to this effect should be inserted earlier.
While for the most part the authors seem comprehensive in their citations, they should consider including reference or discussion of Gupta and coworkers (e.g. DOI: 10.1021/acsami.8b10107) which seems highly related (tuning HIS functionalized gel mechanical properties with choice of metal).
Have the authors tried calculating the effective increase in crosslink density associated with the addition of Zn from the G' data?
Reviewer 2 Report
The manuscript demonstrates the introduction of intermolecular histidine-metal coordination sites into coiled coil (CC) crosslinked star-PEG hydrogels in order to tune the mechanical properties. This work is a continuation of a recently published communication (Nanoscale, 2018, 10, 22725) where the authors introduced two histidine-metal coordination sites into the sequence of the synthetic heterodimer A4B4 and probed force-induced rupture and rheological behavior in the presence of metals. Here, the authors introduce three histidine-metal coordination sites into the A4B4 sequence at the solvent exposed f positions of the heterodimers and are spaced by one heptad to avoid intramolecular interactions. Cysteine residues are added at the N and C terminus of the peptides to facilitate coupling to PEG-maleimide. The PEG-peptide conjugates are mixed to form the CC-crosslinked hydrogel with subsequent addition of metals ions (Ca2+ and Zn2+). The data suggest that the presence of 1:1 Zn2+: histidine increases the shear modulus (G’) by a factor of four and increases the relaxation time by 3 orders of magnitude compared to the hydrogel without metal ions or with Ca2+ ions. Finally, the authors show that the introduction of the second crosslinking mode leads to the formation of Zn2+- crosslinked aggregates which offer the possibility to switch the properties of a hydrogel from viscoelastic to elastic and vice-versa. Overall, I think the paper is technically solid although the structural claims in the schematics are only partially supported by data.
Comments to be considered by the authors are given below.
· The authors may wish to comment on the (in)stability of the Mal-Cys linker used in this work. It has been reported that this linker is susceptible to retro-Michael addition reaction and this may alter the network dynamics observed in the cross-linked polymer systems reported here.
· In Fig. 2A in the circular dichroism spectrum of the Ca2+:His there is a shift on the 208nm minimum peak to a lower wavelength. Why is that?
· In Fig. 2B the Raman spectrum of the Zn2+:His shows a shoulder at 1680 cm-1 which is assigned either to the existence of β-sheet or unordered secondary structure. The authors then claim that the addition of Zn2+ induces the hierarchical assembly of CCs into clusters. Strictly speaking, since the small peak in Fig. 2B cannot be explained, no direct evidence is provided for the clustering of CCs as shown in the cartoons. A structural characterization technique like SAXS would be necessary to claim this.
· What was the rational for choosing Ca2+ as a binding metal? The authors should elaborate more and discuss why the gels with Ca2+ ions have similar rheological behavior to the control gel without metals. Although Zn and other transition metals are discussed in the introduction, Ca is not.
· In Fig. 3 the inset photographs of the various hydrogels are blurry, and they don’t add any additional information in the graph. The authors should provide a better version of the photographs or remove them.
· I am curious why the authors chose to demonstrate recovery of modulus by reversing stepwise from high strain back down to low. Most researchers monitor recovery of modulus under low shear amplitude, and I believe this is more relevant to practical applications than doing the sweep to high strain and then reversing stepwise back to low strain.
· The authors state that their hydrogel formation strategy mimics the self-healing approach of the byssal threads. However, in my opinion it should be stated that their approach is an inspired one, as the natural system is more complex and sophisticated, and does not have CCs.
